# Glucose and Lipid Metabolism Disorders in Adults with Spinal Muscular Atrophy Type 3

**DOI:** 10.3390/diagnostics14182078

**Published:** 2024-09-19

**Authors:** Marija Miletić, Zorica Stević, Svetlana Vujović, Jelena Rakočević, Ana Tomić, Milina Tančić Gajić, Miloš Stojanović, Aleksa Palibrk, Miloš Žarković

**Affiliations:** 1Clinic for Endocrinology, Diabetes and Metabolic Diseases, University Clinical Center of Serbia, 11000 Belgrade, Serbia; prof.svetlana.vujovic@gmail.com (S.V.); mtancicgajic@yahoo.com (M.T.G.); specmedico@gmail.com (M.S.); milos.zarkovic@gmail.com (M.Ž.); 2Faculty of Medicine, University of Belgrade, 11000 Belgrade, Serbia; zstevic@gmail.com (Z.S.); palibrk17@gmail.com (A.P.); 3Clinic of Neurology, University Clinical Center of Serbia, 11000 Belgrade, Serbia; 4Institute of Histology and Embryology “Aleksandar Đ. Kostić”, Faculty of Medicine, University of Belgrade, 11000 Belgrade, Serbia; jelena.rakocevic@med.bg.ac.rs; 5Center for Radiology Imaging-Magnetic Resonance, University Clinical Center of Serbia, 11000 Belgrade, Serbia; anatomic9977@gmail.com

**Keywords:** spinal muscular atrophy type 3, dyslipidemia, insulin resistance, metabolic consequences

## Abstract

Background: Spinal muscular atrophy type 3 (juvenile SMA, Kugelberg–Welander disease) is a genetic disease caused by changes in the survival motor neuron 1 (SMN) gene. However, there is increasing evidence of metabolic abnormalities in SMA patients, such as altered fatty acid metabolism, impaired glucose tolerance, and defects in the functioning of muscle mitochondria. Given that data in the literature are scarce regarding this subject, the purpose of this study was to estimate the prevalence of glucose and lipid metabolism disorders in adult patients with SMA type 3. Methods: We conducted a cross-sectional study of 23 adult patients with SMA type 3 who underwent a comprehensive evaluation, including a physical examination, biochemical analysis, and an oral glucose tolerance test during 2020–2023. Results: At least one lipid abnormality was observed in 60.8% of patients. All four lipid parameters were atypical in 4.3% of patients, three lipid parameters were abnormal in 21.7% of patients, and two lipid parameters were altered in 8.7% patients. A total of 91.3% of SMA3 patients met the HOMA-IR criteria for insulin resistance, with 30.43% having impaired glucose tolerance. None of the patients met the criteria for a diagnosis of overt DM2. Conclusions: The prevalence of dyslipidemia and altered glucose metabolism in our study sets apart the adult population with SMA3 from the general population, confirming a significant interplay between muscle, liver, and adipose tissue. Ensuring metabolic care for aging patients with SMA 3 is crucial, as they are vulnerable to metabolic derangements and cardiovascular risks.

## 1. Introduction

Spinal muscular atrophy (SMA) is a hereditary neuromuscular disease characterized by progressive muscle weakness resulting from degeneration of motor neurons (MN) in the spinal cord [1]. Although SMA is considered a rare disease with an estimated global incidence of about 1:6000 to 1:10,000 live births, SMA remains the second most common autosomal recessive genetic disease and the most common monogenic disorder, causing early infant death [2]. The gene carrier frequency varies from 1 in 38 to 1 in 72 among different ethnic groups, with a pan-ethnic average of 1 in 54 [3,4].

The most common form of SMA is caused by a homozygous deletion or a heterozygous deletion combined with a point mutation on the other allele in the survival motor neuron 1 (SMN1) gene on chromosome 5q and a consequential lack of SMN proteins, causing the degeneration of lower motor neurons [5]. Unlike the SMN1 gene, SMN2 can synthesize only 10% of the full-length SMN [6]. Given that the remaining full-length (FL) SMN2 (SMN2) transcripts can compensate for the SMN1 defect to a limited extent, the severity of the clinical expression of SMA is mitigated by the number of SMN2 copies [7]. However, the correlation of the phenotype and genotype is incomplete. Recent research has highlighted the potential involvement of other cellular mechanisms in modifying the clinical severity of SMA [8].

Conventionally, four main types of SMA (types 1–4) have been distinguished based on the age of onset, maximum motor achievement, and rate of progression [9]. SMA type 1 (infantile, Werdnig–Hoffmann disease) represents the most severe end of the disease spectrum, while SMA type 4 (adult-onset) is the mildest form of the disease. SMA type 2 (intermediate type, Dubowitz disease) usually becomes apparent by the age of 18 months. Patients with this form of the disease may sit unsupported, but are not able to stand independently. SMA type 3 (juvenile, Kugelberg–Welander disease) typically presents after the age of 18 months. Affected patients achieve the ability to walk unaided but lose this ability as the disease progresses. The length of life of patients with SMA3 is mostly unchanged by the presence of the disease [10].

It is still unclear whether the pathogenesis of SMA is caused by a specific pattern or a combination of dysregulated effects of SMN protein deficiency. Cell-autonomous effects due to SMN deficiency are the main causes of motor neuron degeneration; however, this does not fully explain the SMA phenotype and implies that not only are neuronal networks dysregulated but also other non-neuronal cell types are involved in SMA pathology [11,12]. New research extends the pathogenic effect of SMN deficiency beyond the motor neuron to include other cells within and outside the central nervous system. Many peripheral organs and non-neuronal tissues exhibit pathological changes in preclinical SMA models and affected patients [13,14,15].

Furthermore, there is increasing evidence of metabolic abnormalities in SMA patients, such as altered fatty acid metabolism, impaired glucose tolerance, and defects in mitochondrial function in muscle [5,16,17,18].

The purpose of this study was to estimate the prevalence of glucose and lipid metabolism disorders in adult patients with SMA type 3 given the pronounced scarcity of data in the literature on this particular subject. There is an unmet need for an investigation of metabolic status in these patients in order to maximize their quality of life and minimize the consequences of muscle–adipose tissue disbalance.

## 2. Materials and Methods

This study was designed as a cross-sectional study of 23 adult patients with SMA type 3 and was conducted from July 2020 to July 2023 at the Clinic of Endocrinology, Diabetes and Metabolic Diseases, University Clinical Center of Serbia, Belgrade. All adult patients with genetically confirmed SMA type 3 in Serbia at the Clinic of Neurology, University Clinical Center of Serbia, were eligible for the study. The diagnosis of SMA type 3 was made at 12.82 ± 5.11 years. None of the patients started disease-modifying therapy during the study. In the overall cohort, there were 11 (47.8%) men, with an average age of 41.5 ± 14.0 years, and 12 (52.2%) women, with an average age of 39.7 ± 13.1 years; there was no significant age difference (*p* = 0.754). Two women (16.6%) were menopausal. This research was approved by the respective Institutional Ethics Committees (25/VI-3). All study participants provided informed consent.

### 2.1. Patient Data

A detailed medical and family history was obtained from each patient, including the use of medications that may affect blood glucose or lipid levels and a family history of diabetes and dyslipidemia. Anthropometric measurements were performed according to standardized procedures. Height and weight were measured using a digital scale with an accuracy of 1 cm and 1 kg. For immobile patients, the value of the last height measurement while they were able to stand was recorded. The body mass index (BMI) was classified according to a report of the Expert Consultative Group of the World Health Organization (WHO) [19]. An individual was considered underweight if his/her BMI was in the range of 15 to 19.9 kg/m^2^, normal weight if BMI was 20 to 24.9 kg/m^2^, overweight if BMI was 25 to 29.9 kg/m^2^, and obese if BMI was 30 to 35 kg/m^2^ or higher. At the time of endocrinological investigation, 14 (60.9%) patients were wheelchair-dependent and 9 (39.1%) were using ambulatory assistive devices.

### 2.2. Blood Analyses

Biochemical analyses included serum glucose and insulin levels, lipid profile (total cholesterol [TC], high-density lipoprotein cholesterol [HDL-C], non-HDL-C, low-density lipoprotein cholesterol [LDL-C], and triglycerides [TG]), apolipoprotein A 1, apolipoprotein B, and hemoglobin A1c (HbA1c). Blood samples were obtained in the morning at 8 am after a 10- to 12-h overnight fas. Glycemia (mmol/L) was determined by the glucose-oxidase method (Randox, Great Britain), using an auto-analyzer (Beckman, Austria). TC (mmol/L) and TG (mmol/L) were determined by standard enzymatic methods (TC: cholesterol oxidase, Randox, Great Britain; TG: glycerol-3 phosphate oxidase, Randox, Great Britain); HDL (mmol/L) was measured by the direct method (Randox, Great Britain); non-HDL-C were calculated by subtracting HDL-C from TC. LDL-C was determined using the Friedewald formula: LDL-C = TC − (HDL-C + TG/5) [20]. HbA1c was determined via the immunoturbidimetric method using the Architect c8000 analyzer., Abbot Laboratories, Inc. Insulin levels were measured via the chemiluminescence immunoassay (CLIA) method using the Immulite 2000 immunoassay system (Siemens Healthcare, Erlangen, Germany).

Patients with impaired fasting glucose (glucose levels from 5.6 to 6.9 mmol/L [100–125 mg/dL]) or HbA1c between 5.7 and 6.4% were considered to have prediabetes. Fasting glucose concentration ≥ 7 mmol/L (126 mg/dL) or HbA1c ≥ 6.5% were consistent with diabetes [21]. We used the Homeostasis Model Assessment of Insulin Resistance (HOMA-IR) as a surrogate measure of insulin resistance [22]. HOMA-IR was calculated using the following formula: HOMA-IR = (glucose [mmol/L] × insulin [μIU/mL])/22.5.

### 2.3. Oral Glucose Tolerance Test

The OGTT was performed at 8:00 following a 10 to 12 h overnight fast. The participants adhered to their usual unrestricted diet in the days preceding the test. After the insertion of an intravenous cannula, a baseline blood sample was taken for serum glucose and insulin. The patients then took an oral glucose solution containing 1.75 g of glucose per kilogram of body weight (maximum 75 g) within five minutes. The next blood samples for serum glucose and insulin were taken at 30, 60, 90, and 120 min after the glucose drink. Patients with glucose levels between 7.8 and 11.0 mmol/L (140–199 mg/dL) and ≥11.1 mmol/L (200 mg/dL) at 120 min were considered to have impaired glucose tolerance (prediabetes) and diabetes, respectively [22].

### 2.4. Statistical Analysis

Numerical data are expressed as means and standard deviations or as medians with interquartile ranges, where appropriate. Correlation between the variables was assessed using Spearman’s rank correlation coefficient. *p*-values less than 0.05 were considered significant. The IBM SPSS Statistics software version 23.0 for Windows (Armonk, New York, NY, USA) was used for statistical analysis.

## 3. Results

This study included 23 adult patients with SMA type 3. The diagnosis of SMA type 3 was made at 12.82 ± 5.11 years. Eleven patients (47.8%) were men, with an average age of 41.5 ± 14.0 years, and 12 (52.2%) were women, with an average age of 39.7 ± 13.1 years; there was no significant age difference (*p* = 0.754). Demographic and anthropometric data, as well as laboratory parameters, are presented in Table 1.

In the overall cohort, eleven patients (47.8%) were characterized as normal weight, ten (43.5) patients were classified as overweight, and two patients met the criteria for obesity. Among women, three (25%) were overweight, and none was obese. Among men, seven (63.3%) were overweight, and two met the criteria for obesity. None of the patients had been receiving antidiabetic or lipid-lowering drugs. There was a highly statistically significant difference in height and weight between men and women (*p* < 0.001).

At least one lipid abnormality was observed in 14 (60.8%) patients in our cohort. In one patient, all four lipid parameters were atypical, five (21.7%) patients had three abnormal lipid parameters, and two (8.7%) patients had two altered lipid parameters. Seven (30.4%) patients had high LDL cholesterol values, five (21.7%) patients had low HDL values, four (17.3%) patients had high TC values, and five (21.7%) patients had borderline high TC values (Figure 1). The mean Apo B/Apo A ratio was 0.74. The ratio of Apo B to Apo A showed a high risk of coronary heart disease in nine (39.1%) patients, a moderate risk in five patients (21.7%), and a low risk in eight (34.7%) patients.

Women had higher HDL values (*p* = 0.001) and lower Apo B/Apo A ratios than men (*p* = 0.003).

Glucose and insulin levels during the oral glucose tolerance test (OGTT) are shown in Table 2.

Among adult patients with SMA type 3, a total of 21 patients (91.3%) met the HOMA-IR criteria (mean 3.26 ± 1.65) for insulin resistance. Seven patients (30.43%) fulfilled the criteria for impaired glucose tolerance (IGT). None of the patients met the criteria for a diagnosis of overt DM2. Only one patient had a fasting glucose of 6.6 mmol/L, which indicates impaired fasting glucose (IFG). Nine patients (39.1%) had glucose values above 8.60 mmol/L at the 60th minute of the OGTT. Two (8.7%) patients had a family history of type 2 diabetes (Table 3).

The relationship between body mass index (BMI) and HOMA-IR is shown in Figure 2.

A statistically significant correlation between BMI and HOMA-IR was found in our patients (*r* = 0.469, *p* = 0.032).

Line diagrams showing serum glucose and insulin levels during the OGTT in patients with normal glucose levels at the 60th minute (below 8.6 mmol/L) are shown in Figure 3.

Thirteen of fourteen patients (92.1%) with glucose levels lower than 8.6 mmol/L at the 60th minute glucose had normal glucose tolerance.

Line diagrams showing serum glucose and insulin levels during the OGTT in patients with elevated glucose levels at the 60th minute (below 8.6 mmol/L) are presented in Figure 4.

Four of nine patients (44.4%) with glucose levels above 8.6 mmol/L at the 60th minute had normal glucose tolerance.

## 4. Discussion

SMA patients have an imbalance between energy intake and energy expenditure, and altered body composition characterized by reduced lean body mass and increased fat mass. Non-ambulatory but functioning SMA patients are particularly prone to adiposity [23,24,25]. Insulin resistance refers to a reduced biological response of cells and tissues to insulin and is considered to be a causal link between adiposity and type 2 diabetes. Prediabetes is an umbrella term for impaired fasting glucose levels, impaired glucose tolerance (as determined by the OGTT), and HgbA1c levels of 5.7–6.4% [21]. The suppression of endogenous glucose production is maximal at the 60th min of the OGTT and remains suppressed at the same level until the 120th min. The increase in glucose concentration during the OGTT stimulates glucose deposition in peripheral tissues, predominantly in skeletal muscles. Considering that there should not be significant production of endogenous glucose between the 60th min and the 120th min of the OGTT, the drop in glucose concentration after the 60th min of the OGTT primarily reflects the uptake of glucose into peripheral tissues, i.e., skeletal muscles [26].

During a review of the literature, we did not find studies that analyzed glucose tolerance via the OGTT in adults with SMA type 3. In our study, 39.1% of subjects had glucose levels that were higher than 8.6 mmol/L at the 60th minute of the OGTT. Only one patient had impaired fasting glucose (IFG). In IFG, there is a significant IR in the liver and an almost normal insulin sensitivity in the muscles, while in IGT, the opposite situation exists [27], which was present in 30.43% of our patients. Although both conditions are characterized by a reduced early-phase insulin secretion, individuals with IGT had an impaired late-phase insulin secretion and an early rise in glucose concentration at the 30th minute of the OGTT. This rise continued until the 60th minute of the test and remained ≥7.8 mmol/L at the 120th minute of the test. Abdul-Ghani et al. showed that a glucose concentration of ≥8.60 mmol/L at the 60th min of the OGTT in the presence of normal glucose tolerance can identify an increased risk of developing DM2 [28]. 

In our study, four of nine (57.1%) patients with glucose levels above 8.6 mmol/L at the 60th minute had a normal glucose tolerance (NGT). The OGTT analysis of glucose levels at the 60th minute verifies that around 17% of people without glucose intolerance who are at medium and high risk of developing T2DM would benefit from preventive measures [29], which is of great importance for adults with SMA 3. The results from a larger prospective study suggested that a high level of glucose at the 60th minute during the OGTT can serve as a surrogate marker of IR in individuals who have NGT [30]. It is considered that the difference in insulin sensitivity between people with NGT and high glucose levels at the 60th minute during the OGTT and others with NGT is primarily due to larger amounts of visceral and total fat tissue [30], which highlights the higher ratio of adipose tissue to muscle tissue in patients with SMA type 3.

The study of Dahl and Peters analyzed dyslipidemia in SMA patients with an average age of 41.5 years but did not evaluate glucose tolerance [31]. Also, interpretations of the data obtained at that time require caution because the genetic analyses relating to an accurate diagnosis of muscle atrophy were not available. According to our knowledge, studies investigating the association between the SMN gene and diabetes are lacking [32,33,34]. However, studies examining a potential association between the SMN gene and diabetic polyneuropathy have been published [35]. A small pilot study by Davis et al. showed that three of six children with SMA type 2 had impaired glucose tolerance, and five were insulin resistant [17], which was also observed in 53% of patients in a more recent study by Kobayashi et al. [35]. Similar findings were observed for the oral glucose tolerance test in 7 of 15 patients with SMA types 2 and 3, and 29.7% of this cohort met the laboratory criteria for prediabetes [36]. The global prevalence of insulin resistance in adults ranges from 15.5 to 46.5% [37], which strongly contrasts with 91.3% patients with IR in a cohort of adult SMA type 3 patients.

The mean HbA1c in SMA type 3 patients from our cohort was 5.07%. In studies by Brener et al. and Djordjevic et al., HbA1c was around 4.9% [36,38]. In a study by Deguise et al., among 72 patients with primarily SMA types 1 and 2 and an average age of 3.8 years, 57% of patients had a HbA1c below 5% [39]. HbA1c levels are usually within the normal range in the majority of SMA patients studied [38,39]. Our data are in accordance with the data from Brener et al. [38] and Djordjevic et al. [36].

In our cohort, 17.39% of patients had borderline high triglyceride values, and 30.43% of patients had significantly high LDL. Increased TG concentrations are the main source of the production of small dense LDL, which is an independent predictor of cardiovascular diseases [40]. Hypertriglyceridemia results from increased secretion of VLDL-rich TGs and decreased hydrolysis of VLDL and chylomicrons in plasma secondary to decreased lipoprotein lipase (LPL) activity. Insulin resistance is one of the factors that inhibits the activity of LPL [41].

In our patient cohort, 18 patients (78.26%) had elevated non-HDL levels, which were above 2.6 mmol/L. The mean non-HDL value in our subjects was 3.68 + 1.22 mmol/L. Six of these eighteen patients (30%) also had normal LDL cholesterol values, which would have remained unrecognized with standard lipid analysis. Non-HDL levels represent the difference between total cholesterol and HDL cholesterol levels [42]. Non-HDL-C has been used as a risk assessment tool after its inclusion as a secondary target of treatment by the National Cholesterol Education Program for Adults (NCEP ATP-III), especially in patients with hypertriglyceridemia, diabetes mellitus (DM), obesity, or low LDL-C [43].

Among our patients with SMA type 3, 21.73% had low HDL values, and all met the criteria for insulin resistance. Low HDL levels increase the risk of coronary disease if they exist in the presence of IR (HR of 2.83 compared to people without IR) [44], which was the case in 91.3% of our patients.

The median ApoB/Apo A ratio in adults with SMA was 0.74 ± 0.31. Five patients (21.7%) were in the high-risk range, seven (30.4%) were in the moderate-risk range, and five (21.7%) were in the low-risk range. Women had a significantly lower Apo B/Apo A ratio than men. During a review of the literature, we did not find studies that analyzed the ApoB/Apo A1 ratio in patients with SMA type 3. A higher Apo B/Apo A ratio corresponds with a greater probability of induction of endothelial dysfunction and the emergence of the atherogenic process.

The percentage of patients with at least one lipid abnormality was 60.8%, which is certainly due to the aging process. A recent study by Djordjevic et al. in adolescent SMA type 2 and type 3 patients found that 30% of patients had at least one lipid level abnormality. Sixteen patients (43.2%) had at least one abnormal finding (prediabetes, dyslipidemia, or hepatic steatosis) [36]. A study by Deguise et al. reported that 37.5% of patients with primarily SMA types 1 and 2 and an average age of 3.8 years had an increased risk of developing dyslipidemia, and there was evidence of hepatic steatosis in their pathological specimens. Deguise et al. also reported a higher prevalence of abnormal blood lipid levels in children with SMA than in the general pediatric population [45]. During 2015–2018, the prevalence of high total cholesterol among adults aged ≥20 years was 11.4%. Adults aged 40–59 years (15.7%) had the highest prevalence of high total cholesterol, and those aged 20–39 years (7.5%) had the lowest prevalence [46].

Dysregulated lipid metabolism is the first and most studied nutritional problem in SMA [47,48]. Abnormal levels of fatty acid oxidation metabolites such as esterified carnitine, which can cause dicarboxylic aciduria, were first reported in several studies of patients with severe SMA [48,49,50]. It has recently been found that defects in fatty acid transport and mitochondrial beta oxidation may also contribute to muscle atrophy in patients with a severe SMA phenotype [51]. However, the exact mechanism of this abnormality of lipid metabolism in SMA is still unclear, but it is assumed to be related to the absence of the SMN gene product, defects in neighboring genes, or the eventual loss of neuronal trophic factor [51,52]. In particular, in some patients with SMA, metabolic dysregulations may appear even before the first neuromuscular signs of the disease [53].

## 5. Conclusions

The presence of dyslipidemia and insulin resistance in our cohort of adult patients with SMA type 3 sets them apart from the general population, confirming a significant interplay between muscle, liver, and adipose tissue. This interaction is much more complex than a simple mechanistic link.

Ensuring metabolic care for aging patients with SMA type 3 is crucial, as they are vulnerable to metabolic derangements and cardiovascular risks. Given that their life expectancy aligns with the general population, it is essential to provide them with adequate care with an emphasis on adequate nutritive support, encourage them to engage in physical activity in accordance with their present physical abilities, and, where appropriate, treat them with insulin sensitizers. Regular metabolic screening and close monitoring throughout adulthood are imperative to maximize their quality of life and mitigate the impact of muscle and adipose tissue imbalance.

A limitation of our study is that it consisted of a small cohort of adult patients with SMA type 3. Further investigations with larger cohorts are needed to confirm the impact of the present findings.

## Figures and Tables

**Figure 1 diagnostics-14-02078-f001:**
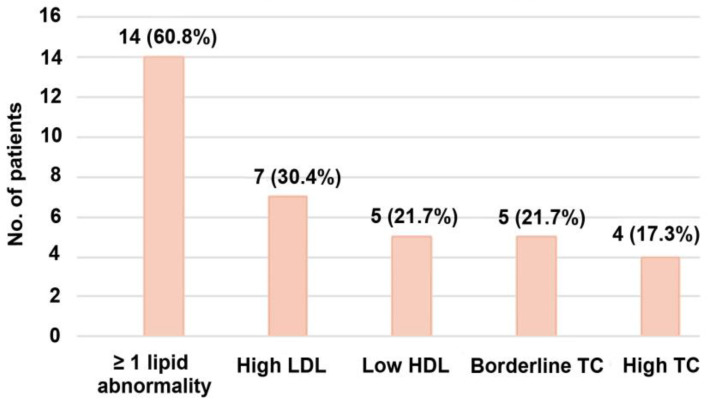
The prevalence of dyslipidemia expressed in lipid fractions. LDL—low-density lipoprotein, HDL—high-density lipoprotein, TC—total cholesterol.

**Figure 2 diagnostics-14-02078-f002:**
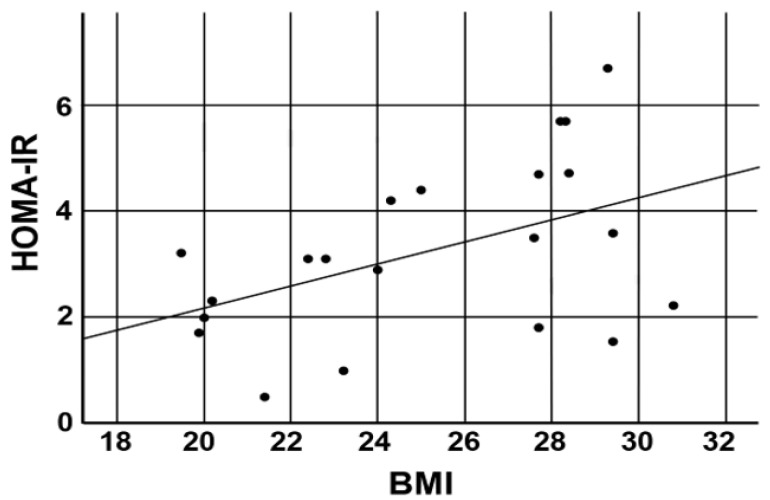
Scatter plot showing the relationship between body mass index (BMI) and HOMA-IR.

**Figure 3 diagnostics-14-02078-f003:**
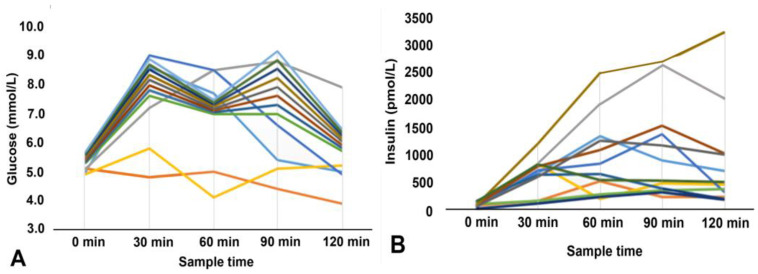
Glucose (**A**) and insulin levels (**B**) in patients with glucose levels below 8.6 mm/L at the 60th minute.

**Figure 4 diagnostics-14-02078-f004:**
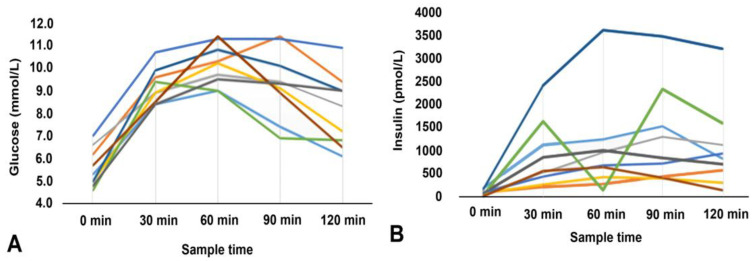
Glucose (**A**) and insulin levels (**B**) in patients with glucose levels >8.6 mm/L at the 60th minute.

**Table 1 diagnostics-14-02078-t001:** Demographic and anthropometric data of 23 patients with SMA type 3.

Patient Characteristics (*n* = 23)	Overall	Male (*n* = 11; 47.8%)	Female (*n* = 12; 52.2%)	*p*
Age	40.6 ± 13.2	41.5 ± 14.0	39.7 ± 13.1	0.754
Weight (kg)	72.6 ± 15.9	74.8 ± 13.3	61.4 ± 7.8	<0.001
Height (cm)	168.7 ± 8.6	173.0 (170.0–175.0)	161.5 (160.0–168.7)	<0.001
Body Fat (%)	45.82 ± 8.53	40.49 ± 10.44	47.15 ± 4.53	0.098
Parameters of glucose and lipid metabolism				
Glucose (mmol/L)	5.02 ± 0.57	5.11 ± 0.75	4.94 ± 0.34	0.491
HbA1c (%)	5.07 ± 0.36	5.10 ± 0.22	5.04 ± 0.46	0.740
TC (mmol/L)	5.04 ± 1.08	5.25 ± 1.14	4.84 ± 1.02	0.370
HDL (mmol/L)	1.36 ± 0.37	1.11 ± 0.29	1.59 ± 0.27	0.001
LDL (mmol/L)	3.03 ± 1.00	3.33 ± 1.00	2.75 ± 0.96	0.166
Non-HDL (mmol/L)	3.68 ± 1.22	4.14 ± 1.33	3.25 ± 0.98	0.080
TG (mmol/L)	1.10 (0.90–1.46)	1.10 (1.00–1.90)	1.00 (0.62–1.27)	0.078
Apo A (g/L)	1.47 ± 0.23	1.35 ± 0.21	1.60 ± 0.18	0.008
Apo B (g/L)	1.04 ± 0.34	1.20 ± 0.35	0.88 ± 0.25	0.025
Apo B/Apo A	0.74 ± 0.31	0.93 ± 0.32	0.55 ± 0.16	0.003

Data are expressed as numbers and frequencies (%), mean ± standard deviation, or median (interquartile range); TC—total cholesterol; HDL—high-density lipoprotein; LDL—low-density lipoprotein; TG—triglycerides; Apo A—apolipoprotein A; Apo B—apolipoprotein B; Apo B/Apo A—ratio of apolipoproteins A and B.

**Table 2 diagnostics-14-02078-t002:** Glucose and insulin levels during OGTT in patients with SMA type 3.

Glucose sample (minutes)	Glucose (mmol/L)
0	5.2 ± 0.69
30	7.81 ± 1.72
60	8.14 ± 2.16
90	7.73 ± 2.06
120	6.86 ± 1.86
Insulin sample (minutes)	Insulin (pmol/L)
0	95.0 (59.2–114.4)
30	668.8 (370.4–858.1)
60	706.3 (376.6–1257.0)
90	863.9 (430.1–1528.9)
120	705.6 (328.0–1080.2)
HOMA Index	3.26 ± 1.65

Data are expressed as mean ± standard deviation or median (interquartile range).

**Table 3 diagnostics-14-02078-t003:** Clinical findings for impaired glucose metabolism.

Positive Family History of DM Type 2	2 (8.7%)
Insulin resistance based on HOMA-IR	21 (91.3%)
Impaired glucose tolerance	7 (30.43%)
Impaired fasting glucose	1 (4.35%)
Glucose values above 8.60 mmol/L at the 60th minute of the OGTT	9 (39.1%)

DM—diabetes mellitus; HOMA-IR—Homa Insulin Resistance Index; OGTT—oral glucose tolerance test.

## Data Availability

The data presented in this study are available on request from the corresponding author. The data are not publicly available due to planned extension of enrollment of patients and formation of even larger cohort. Additionally, these data will be used as a amatrial for PhD thesis of one of the authors and therefore require presesrvation untill the presentation of the PhD thesis results.

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
