# Peer review of "Glucose and Lipid Metabolism Disorders in Adults with Spinal Muscular Atrophy Type 3"

_diagnostics, 2024, doi:10.3390/diagnostics14182078_

Round 1

Reviewer 1 Report

Comments and Suggestions for Authors

The work “Glucose and lipid metabolism disorders in adults with spinal muscular atrophy type 3” by Dr. Miletić and colleagues is of great interest to the scientific community working in the field of neurology of inherited neuromuscular diseases. In the study, the authors comprehensively investigated the impairment of glucose and lipid metabolism in patients with rare neuromuscular disorder - spinal muscular atrophy type 3. The article is written in good language and contains well-designed graphics and illustrations. During the reading, the reviewer had several questions and comments, as follows:

MAJOR

Despite the good design, well-presented results, and correct conclusions, the manuscript lacks a crucial table containing information on the clinical (neurological) symptoms of the patients, such as their walking status and muscle strength parameters, as well as their genetic characteristics, including the SMN2 gene copy number and the method by which they were diagnosed.

MINOR

It would also be useful to include information on any therapeutic interventions that were administered to the patients in the study cohort after the study's completion to improve their metabolic status.

Additionally, it would be beneficial to provide some clinical perspectives for SMA type 3 patients with regard to their impaired metabolic status.

Reviewer 2 Report

Comments and Suggestions for Authors

Dear Authors,

Thank you for the opportunity to review this important article. 

My main concern is that the study does not have a control group of healthy individuals. Ordinarily such a group should be present.

Other comments in order of appearance in the text (not in order of importance):

1. keywords should be different from those in the title. All words of the title are keywords.

2. Please explain why non-standard BMI categories have been used, whereas in the letter a reference to the standard breakdown is given.

3. In the Material section from the beginning it should state how many men how many women were studied, the age of each sex (and SD). 

4. in the Results section, all results should be given separately for males and females. The standard deviation of age for women is large, which means that there are also women of menopausal age among the subjects, which may be related to the level of biological parameters analysed (e.g. BMI).

5. in Figure 1, % and not n values should be given and the description in the text should also refer to % and not n (vers 147-154).

6) The age of the subjects is repeated in verses 132 and 135.

7. The limitations of the material and the study conducted should definitely be stated.

Kind regards,

reviewer

Round 2

Reviewer 1 Report

Comments and Suggestions for Authors

The authors successfully answered the questions.